# Quantum test of the equivalence principle for atoms in coherent superposition of internal energy states

G. Rosi[1], G. D'Amico[1], L. Cacciapuoti[2], F. Sorrentino[3], M. Prevedelli[4], M. Zych[5], Č. Brukner[6,7] & G.M. Tino[1]

The Einstein equivalence principle (EEP) has a central role in the understanding of gravity and space–time. In its weak form, or weak equivalence principle (WEP), it directly implies equivalence between inertial and gravitational mass. Verifying this principle in a regime where the relevant properties of the test body must be described by quantum theory has profound implications. Here we report on a novel WEP test for atoms: a Bragg atom interferometer in a gravity gradiometer configuration compares the free fall of rubidium atoms prepared in two hyperfine states and in their coherent superposition. The use of the superposition state allows testing genuine quantum aspects of EEP with no classical analogue, which have remained completely unexplored so far. In addition, we measure the Eötvös ratio of atoms in two hyperfine levels with relative uncertainty in the low $10^{-9}$, improving previous results by almost two orders of magnitude.

[1] Dipartimento di Fisica e Astronomia and LENS, Università di Firenze—INFN Sezione di Firenze, Via Sansone 1, Sesto Fiorentino 50019, Italy. [2] European Space Agency, Keplerlaan 1—P.O. Box 299, Noordwijk ZH 2200 AG, The Netherlands. [3] INFN Sezione di Genova, Via Dodecaneso 33, Genova 16146, Italy. [4] Dipartimento di Fisica e Astronomia, Università di Bologna, Via Berti-Pichat 6/2, Bologna 40126, Italy. [5] Centre for Engineered Quantum Systems, School of Mathematics and Physics, The University of Queensland, St Lucia, Queensland 4072, Australia. [6] Faculty of Physics, University of Vienna, Boltzmanngasse 5, Vienna 1090, Austria. [7] Institute for Quantum Optics and Quantum Information, Austrian Academy of Sciences, Boltzmanngasse 3, Vienna 1090, Austria. Correspondence and requests for materials should be addressed to G.M.T. (email: guglielmo.tino@unifi.it).

Several experiments have been performed so far in the attempt to detect weak equivalence principle (WEP) violations and unveil new physics beyond general relativity and the standard model[1]. They compare the free-fall accelerations $a_A$ and $a_B$ of different test bodies, A and B. The relative differential acceleration $\eta_{A-B}$ provides the Eötvös ratio, which is a measure of the WEP violation:

$$\eta_{A-B} = 2 \times \frac{|a_A - a_B|}{|a_A + a_B|} = 2 \times \frac{\left|(m_i/m_g)_A - (m_i/m_g)_B\right|}{\left|(m_i/m_g)_A + (m_i/m_g)_B\right|}, \quad (1)$$

where $m_i$ and $m_g$ denote the inertial and gravitational mass. The most stringent bounds on $\eta$ are today provided by torsion balance tests[2]. They measure the differential acceleration of macroscopic objects of different composition to better than $2 \times 10^{-13}$. Similar accuracy levels are provided by laser ranging experiments tracking the orbital motion of the Moon[3]. In space, the MICROSCOPE mission[4] is currently using a differential accelerometer to compare the free fall of a Ti versus Pt:Rh test body aiming at $1 \times 10^{-15}$. The rapid development of atom interferometry[5] is now providing instruments for testing WEP at the atomic level, based on different schemes of preparation, control and measurement of the probe masses. After the first atom interferometry test by Fray et al.[6], several experiments have recently compared the free fall of different atoms: $^{85}$Rb versus $^{87}$Rb[7,8] and $^{39}$K versus $^{87}$Rb[9], the bosonic $^{88}$Sr versus the fermionic $^{87}$Sr[10] and atoms in different spin orientations[10,11]. The accuracy of these measurements, now in the $10^{-7}$ to $10^{-8}$ range, is expected to improve by several orders of magnitude in the near future owing to the rapid progress of atom-optical elements based on multiphoton momentum transfer[12,13] and of large-scale facilities providing a few seconds of free fall during the interferometer sequence[14,15]. Experiments testing the free fall of antihydrogen are in progress[16,17]. Finally, STE-QUEST is proposing a WEP test to $1 \times 10^{-15}$ using a differential atom interferometer in space[18]. Beyond distinguishing general relativity from other gravitational theories, experimental tests of Einstein equivalence principle (EEP) are of high interest as its violations are a common low-energy prediction of various quantum gravity frameworks, despite their disparate motivations and mathematical formalisms[19–24].

The EEP requires equivalence of the total rest mass–energy of a body, the mass–energy that constitutes its inertia, and the mass–energy that constitutes its weight. In classical physics, for testing EEP it suffices to compare the values of the mass–energies that are treated as classical variables. In quantum mechanics, internal energy is given by a Hamiltonian operator describing the dynamics of internal degrees of freedom that contributes to the total mass. Note that a general state of the internal energy can involve superpositions of states with different internal energy eigenvalues. Hence, one has to introduce a quantum formulation of EEP that states equivalence between the rest, inertial and gravitational mass–energy quantum operators[25,26]. So far performed experimental tests of EEP are only sensitive to the diagonal elements of the mass–energy operators and hence they can be characterized as tests of the classical EEP. To probe the validity of the quantum EEP one needs to additionally test equivalence between the off-diagonal elements of the operators, which necessarily involves superpositions of states with different energies[27].

In the following, we present the results of our test of the quantum WEP. An atom interferometer compares the free-fall acceleration of rubidium atoms prepared in the two hyperfine levels of the internal energy ground state and in their coherent superposition. Based on first principles, we introduce a quantum formulation of EEP to interpret the results. Within this framework, our measurements provide constraints on the off-diagonal elements of the mass–energy operators. At the same time, we significantly improve current limits of WEP tests performed on rubidium atoms prepared in different eigenstates of the internal energy.

## Results

**The theoretical framework.** According to the mass–energy equivalence, we introduce the mass–energy operators

$$\hat{M}_\alpha = m_\alpha \hat{I} + \frac{\hat{H}_\alpha}{c^2}, \quad (2)$$

with $\alpha = i, g$. Here $\hat{H}_i$ and $\hat{H}_g$ are the contributions of the internal energy to the inertial and gravitational mass, respectively. The quantum formulation of WEP requires $\hat{M}_i = \hat{M}_g$. In a quantum test theory incorporating WEP violations, $\hat{M}_i \neq \hat{M}_g$ and the centre-of-motion acceleration is $\hat{a} = \hat{M}_g \hat{M}_i^{-1} g$, where $g$ is the strength of the local gravitational field. Starting from equation (2), it can be shown that $\hat{M}_g \hat{M}_i^{-1}$ can be represented, to lowest order in $1/c^2$, by a Hermitian operator. In the subspace spanned by the eigenstates $|1\rangle$ and $|2\rangle$ of the internal energy operator $\hat{H}_i$,

$$\hat{M}_g \hat{M}_i^{-1} \approx \begin{pmatrix} r_1 & r \\ r^* & r_2 \end{pmatrix}, \quad (3)$$

where $r = |r|e^{i\varphi_r}$ and $r^*$ is its complex conjugate. The classical WEP is valid if $r_1 = r_2 = 1$, whereas the quantum WEP holds if $r = 0$ in addition to that. The off-diagonal element $r$ introduces a coupling between the two energy eigenstates that could be measured by detecting the relative population of the $|1\rangle$ and $|2\rangle$ state before and after the free-fall experiment. However, such an approach would lead to a quite poor accuracy considering that the probability for such a transition is at least of order $r^2$, while the stability of relative atom number measurements is typically not better than $10^{-3}$. On the contrary, $r$ can be measured by interfering atoms in a coherent superposition of the two energy eigenstates, thus providing a much stringent bound on this parameter.

In our experiment, atom interferometry is used to compare the free fall of laser-cooled $^{87}$Rb samples prepared in the $|1\rangle = |F=1, m_F=0\rangle$ and $|2\rangle = |F=2, m_F=0\rangle$ hyperfine levels of the ground state and in their coherent superposition $|s\rangle = (|1\rangle + e^{i\gamma}|2\rangle)/\sqrt{2}$; here $\gamma$ is a random phase with standard deviation $\sigma \gg 2\pi$, which cannot be controlled from one measurement to the next. The instrument is sensitive to (see Methods):

$$a_1 = g\langle 1|\hat{M}_g \hat{M}_i^{-1}|1\rangle = g r_1, \quad (4)$$

$$a_2 = g\langle 2|\hat{M}_g \hat{M}_i^{-1}|2\rangle = g r_2, \quad (5)$$

$$a_s = g\langle s|\hat{M}_g \hat{M}_i^{-1}|s\rangle = g\left[\frac{r_1 + r_2}{2} + |r|\cos(\varphi_r + \gamma)\right]. \quad (6)$$

Within this framework, a WEP violation introduced by the diagonal elements $r_1$ and $r_2$ would emerge as a non-zero differential acceleration proportional to $r_1 - r_2$; a WEP violation introduced by the off-diagonal element $r$ would manifest itself as an excess of phase noise with zero average on the acceleration measurements, due to randomization of $\gamma$.

**The experiment.** Our instrument is an atomic gravity gradiometer based on a third-order Bragg-pulse interferometer (see Methods). The two Mach–Zehnder interferometers are aligned along the vertical direction, separated by a distance of $\sim 30$ cm (see Fig. 1)[28,29]. The gravity gradiometer can operate in three different configurations (see Methods): with both atomic clouds in the $|1\rangle$ state ($1-1$ configuration); with the upper cloud

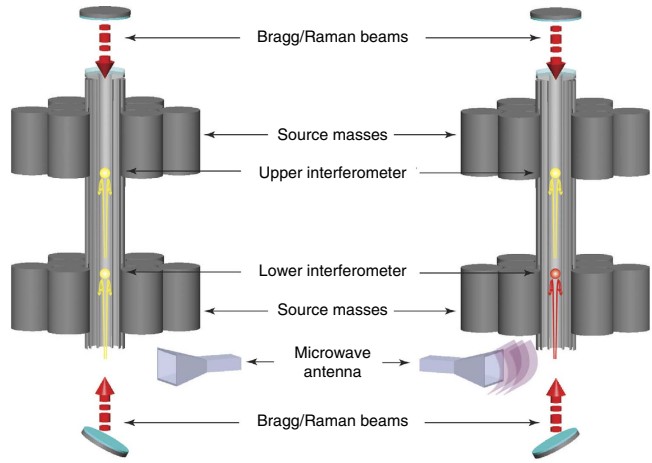

**Figure 1 | Schematic of the experiment.** Two laser-cooled clouds of $^{87}$Rb atoms are launched vertically and prepared in the $|1\rangle$ internal state by velocity-selective Raman pulses. Before the interferometric sequence, a microwave pulse transfers the lower atomic cloud in any superposition of the $|1\rangle$ and $|2\rangle$ states. The Bragg interferometer simultaneously interrogates both atomic clouds, measuring the acceleration of the lower cloud with respect to the upper cloud, which is used as a common reference. The external source masses are positioned to maximize the gravity gradient and optimize the extraction of the differential acceleration from the measurements.

in the $|1\rangle$ state and the lower cloud in the $|2\rangle$ state ($1-2$ configuration); with the upper cloud in the $|1\rangle$ state and the lower cloud in the superposition state $|s\rangle$ ($1-s$ configuration). The upper cloud is then used as a common reference to measure the acceleration experienced by the lower cloud. A key aspect of our WEP test is that the same Bragg lasers are used to simultaneously probe the two hyperfine states $|1\rangle$ and $|2\rangle$ on two identical atom interferometers acting on orthogonal internal states, the first based on red-detuned Bragg transitions, the second on blue-detuned ones. The detuning of the Bragg lasers with respect to the $5^2S_{1/2}|F=2\rangle \to 5^2P_{3/2}|F'=3\rangle$ transition is defined by the condition

$$\Omega_e^{F=1} = \Omega_e^{F=2}, \tag{7}$$

where $\Omega_e^{F=1,2}$ is the effective Rabi frequency for the two-photon Bragg transition from the $F=1, 2$ hyperfine level of the ground state. With Bragg lasers of equal intensity, a magic detuning of 3.1816 GHz can be calculated from equation (7), based on the frequency difference between the $^{87}$Rb hyperfine levels and the dipole matrix elements for $\sigma^+ - \sigma^-$ transitions. From the differential phase shifts $\Phi_{1-1}$, $\Phi_{1-s}$ and $\Phi_{1-2}$ measured for the three gravity gradiometer configurations (see Fig. 2), we can evaluate the differential accelerations of the lower atomic clouds when prepared in the superposition state and in the $|2\rangle$ state with respect to the $|1\rangle$ state: $\delta g_{1-s} \propto (\Phi_{1-1} - \Phi_{1-s})$ and $\delta g_{1-2} \propto (\Phi_{1-1} - \Phi_{1-2})$.

A budget of the systematic uncertainties affecting the differential acceleration measurements is presented in Table 1 and further discussed in Methods. In order of importance, the major error contributions are: the AC Stark shift due to the intensity inhomogeneities induced by diffraction effects on the Bragg beams; the second-order Zeeman effect due to magnetic field inhomogeneities; the uncertainty on the noise affecting experimental data used in the Bayesian analysis to extract the differential phase of the gravity gradiometer. We do not correct our results for any systematic biases, as they are negligible compared to the corresponding uncertainties. On the contrary, systematics contribute a significant error on the differential

acceleration measurements, more than one order of magnitude larger than our statistical uncertainty.

In a first experiment, we use the same detection channel to measure the normalized population in the two momentum states for both the $F=1$ and $F=2$ Bragg interferometers. Atoms are excited on both the $F=1 \to F'$ and $F=2 \to F'$ transitions by the detection lasers and counted by measuring the light-induced fluorescence emission. In this way, we avoid systematic effects arising from asymmetries in the detection channels. Two data sets of 4,320 points (1.9 s per point) are collected by periodically reversing the direction of the Bragg lasers wavevectors and alternating different gradiometer configurations during the data acquisition: $1-1$ and $1-s$ for the test involving the quantum superposition of $|1\rangle$ and $|2\rangle$ states; $1-1$ and $1-2$ for the test on the two eigenstates of internal energy. Figure 2 shows typical data plots together with the best fitting ellipses (see Methods). After correcting for the systematic shifts (see Table 1), we obtain the Eötvös ratios $\eta_{1-2} = (1.4 \pm 2.8) \times 10^{-9}$ and $\eta_{1-s} = (3.3 \pm 2.9) \times 10^{-9}$. Both values provide a direct measurement of $r_1 - r_2$. More importantly, by attributing all the phase noise observed on the $1-s$ ellipse to a WEP violation, we can establish an upper limit to $|r|$ that we estimate to $5 \times 10^{-8}$ (see Methods).

In the second experiment, we operate our gravity gradiometer in the $1-s$ configuration and measure the atomic population in the $F=1$ and $F=2$ states simultaneously by using two independent state-selective detection channels. A total of 4,320 data points are collected for the two opposite directions of the Bragg lasers wavevectors. Figure 3 shows a three-dimensional (3D) plot of the measurements collected at the three conjugated gravity gradiometers, together with the ellipse best fitting the experimental data[30]. From the measurement of the differential phase shifts, we extract the Eötvös ratio $\eta_{1-2}$ within the quantum superposition of the $|1\rangle$ and $|2\rangle$ internal states. In this case, the phase shift introduced by the asymmetry in the two channels used to detect $F=1$ and $F=2$ atoms must be evaluated. To this purpose, we compare the differential phase $\Phi_{1-2}$ measured in the $1-2$ gradiometer configuration when counting $F=2$ atoms via both the first and the second detection channel. After correcting for this effect, which amounts to $(38 \pm 3)$ mrad, we obtain an Eötvös ratio $\eta_{1-2} = (1.0 \pm 1.4) \times 10^{-9}$. Also in this case, the phase noise affecting the ellipse can be used to establish an upper limit on $|r|$ that we evaluate to $5 \times 10^{-8}$.

Our measurements provide a test of the quantum WEP revealing no violation at the level of a few parts in $10^8$. In addition, we improve by almost two orders of magnitude the results of classical WEP tests performed on atoms in different eigenstates of the internal energy[6] by measuring the corresponding Eötvös parameter to one part in $10^9$. Our uncertainty is presently limited by the AC Stark effect due to the intensity gradients of the Bragg beams. In future experiments, a higher power and suitably shaped laser beam together with a light shift compensation scheme[14] can be implemented to reduce this error source by more than one order of magnitude, possibly pushing the test to the $10^{-10}$ accuracy level and beyond. Assuming that WEP violations increase with the energy difference between the internal levels[20], it would be advantageous to use states with an energy gap larger than hyperfine splitting. A feasible aim for near-future experiments would then be optically separated levels, for example, as in strontium[31].

## Methods
**The Bragg-pulse atom interferometer.** In a Bragg-pulse interferometer, atoms coherently interact with two counter-propagating laser beams on a $2n$-photon transition between different momentum states without changing their internal energy. At the $n$th Bragg diffraction order, the two momentum states are separated

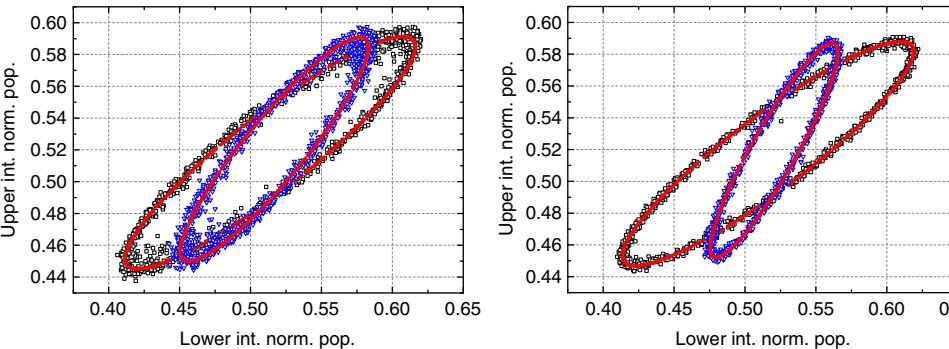

**Figure 2 | Experimental data from the Bragg gravity gradiometers.** (left) With both atomic samples prepared in the $F=1$ state ($1-1$ configuration, black squares), and with the upper sample in $F=1$ and the lower sample in a coherent superposition of the two hyperfine states ($1-s$ configuration, blue triangles). (right) With both atomic samples prepared in the $F=1$ state ($1-1$ configuration, black squares), and with the upper sample in $F=1$ and the lower sample in $F=2$ ($1-2$ configuration, blue triangles). The ellipses best fitting the experimental data are also shown (red lines). The loss of contrast observed on the $|2\rangle$ interferometer can be attributed to the defocusing effect experienced by the atoms when interrogated by the blue-detuned Bragg lasers.

## Table 1 | Measurement systematics.

| Effect | Uncertainty on $\delta g/g(\times 10^{-9})$ |
| --- | --- |
| Second order Zeeman shift | 0.6 |
| AC Stark shift | 2.6 |
| Ellipse fitting | 0.3 |
| Other effects | $<0.1$ |

Main error contributions affecting the differential acceleration measurement.

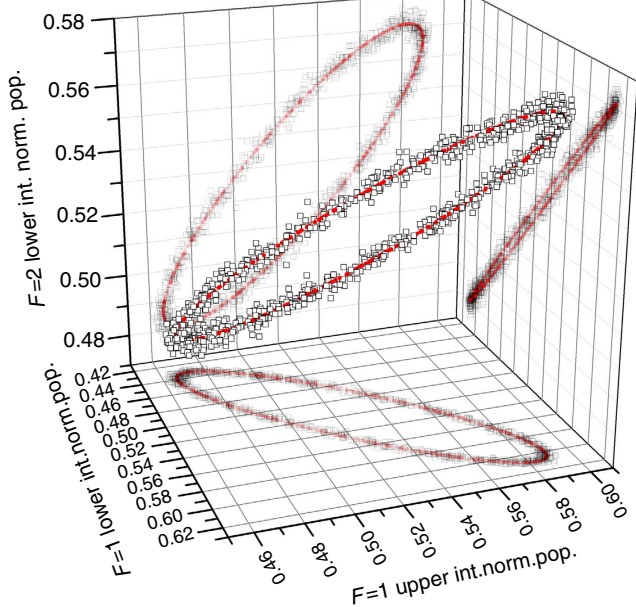

**Figure 3 | 3D plot of the superposition state data.** 3D Lissajous figure obtained by plotting the output signal of the lower $F=2$ atom interferometer as a function of the output signals of the $F=1$ interferometers at the upper and lower clouds (black squares). The 3D ellipse best fitting the data (red line) and the orthogonal projections on the three Cartesian planes are also shown.

by $2n\hbar k$, corresponding to a variation of the kinetic energy of $4n^2\hbar\omega_r$ for an atom initially at rest; here $\mathbf{k}=\mathbf{k_1}\approx-\mathbf{k_2}$ is the wavevector of the counter-propagating Bragg lasers and $\omega_r=\hbar k^2/(2m)$ the corresponding recoil frequency for an atom of mass $m$. The resonance condition is then established by the relationship $\omega_2-\omega_1=4n\omega_r$, where $\omega_1$ and $\omega_2$ are the frequencies of the two Bragg lasers.

Bragg diffraction can be used to implement multiphoton beam splitters and mirrors in a Mach–Zehnder interferometer and improve its sensitivity. A $\pi/2-\pi-\pi/2$ pulse sequence coherently splits, reflects and recombines the atomic wavefunctions generating the interference effects that can be read by detecting the normalized population in the two momentum states $|0\rangle$ and $|2n\hbar k\rangle$. In a vertical configuration, the atomic wavefunction components propagating along the two spatially separated arms of the interferometer acquire a phase difference $\Phi=n(2kgT^2+\phi_L)$ depending on the local acceleration of gravity, where $T$ is the time interval between the central mirror pulse and the two beam splitter pulses and $\phi_L$ the phase contribution of the Bragg lasers[32]. Our atom interferometer operates at the $n=3$ Bragg diffraction order, corresponding to $6\hbar k$ of total momentum transfer between the atoms and the radiation field. The temporal intensity profile of the Bragg pulses is Gaussian, with 24 μs full-width at half-maximum. The $\pi/2$ and $\pi$ pulses of the interferometer sequence are obtained by appropriately tuning the overall power. The total duration of the $\pi/2-\pi-\pi/2$ Bragg-pulse sequence is $2T=160$ ms.

**The gravity gradiometer.** Atomic samples are loaded from a two-dimensional magneto-optical trap into a 3D magneto-optical trap. A moving optical molasses accelerates the atoms upwards at a temperature of $\sim4\,\mu K$. We use the juggling technique to launch the two atomic samples in rapid sequence and separate them by a distance of $\sim30$ cm. Immediately after the launch, a series of velocity-selective Raman pulses prepares $10^5$ atoms into the magnetically insensitive $|F=1,m_F=0\rangle$ sublevel within a narrow vertical velocity distribution of $\sim0.16v_r$ at full-width at half-maximum, where $v_r=\hbar k/m=5.8$ mm s$^{-1}$ is the recoil velocity for the rubidium D2 line. Before the Bragg interferometer, the lower cloud can be prepared in each of the three states $|1\rangle$, $|2\rangle$ and $|s\rangle$ using a microwave pulse resonant with the $|F=1,m_F=0\rangle\rightarrow|F=2,m_F=0\rangle$ magnetic dipole transition. The overall interferometer sequence takes place at the centre of a magnetically shielded vertical tube surrounded by a well-characterized set of source masses[33]. Their positions are accurately tuned to maximize the gravity gradient experienced by the atoms and reach optimal conditions to extract the differential acceleration from the elliptical fit on the gradiometer data points[34] (see below). Atoms are simultaneously interrogated by the same Bragg pulses with a $\pi/2-\pi-\pi/2$ sequence. This configuration is particularly interesting when performing differential acceleration measurements to high precision[29,30]. Indeed, any mechanical vibration at the measurement platform, which manifests itself as common-mode phase noise at the two conjugated atom interferometers, is efficiently rejected.

**Phase shift calculation in the atom interferometer.** The atomic ensemble at the input of the atom interferometer can be prepared in the eigenstates $|1\rangle$, $|2\rangle$ and in their coherent superposition $|s\rangle$, as per the notation in the main text.

During the interferometric sequence, the initial state evolves under a unitary operator $\hat{U}$ into the corresponding output state. The unitary operator $\hat{U}$ accounts for the interaction with the Bragg lasers and for the effects of gravity, including WEP violating terms. The interaction term introduced by the Bragg lasers only couples two different momentum states of the same energy level. At the magic detuning, the coupling coefficients of the Bragg transitions on the $|1\rangle$ and $|2\rangle$ states are the same.

For a particle in a homogeneous gravitational field, the test Hamiltonian incorporating the quantum WEP formulation can be written as

$$\mathcal{H}=\hat{M}_i c^2+\hat{M}_i^{-1}\frac{\hat{p}^2}{2}+g\left(\hat{M}_g\hat{M}_i^{-1}\right)\hat{M}_i\hat{z}, \qquad (8)$$

where $\hat{p}$ and $\hat{z}$ are the atomic momentum and position operators in the direction of

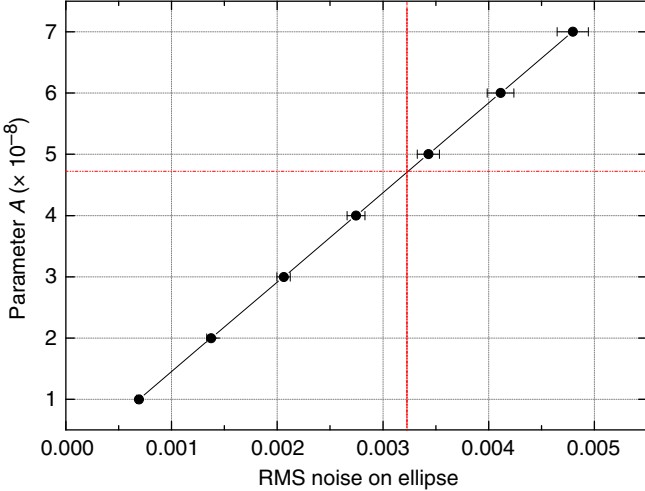

**Figure 4 | Upper limit on |r|.** Parameter $A$ as a function of the RMS noise measured on simulated data points with respect to the best-fitting ellipse. Our ellipses show an RMS noise of 0.0032, corresponding to an upper bound on $|r|$ of $5 \times 10^{-8}$. The error bars represent the standard error on the RMS: $\sigma_{RMS} = RMS/\sqrt{N}$, where $N$ is the number of data points on our ellipses.

the local gravitational field, $g$ is the strength of the local gravitational field and the mass operator $\hat{M}_g \hat{M}_i^{-1}$ is parameterized as in equation (3). Importantly, in the presence of quantum WEP violations $r \neq 0$ and the above Hamiltonian introduces a coupling between different internal energy levels. Relativistic corrections are negligible, thus $\hat{M}_g \hat{M}_i^{-1}$ does not couple atoms in different momentum states.

In the regime of homogeneous gravity, when evaluating transition amplitudes in the atom interferometer, the gravitational potential can be treated as a perturbation of the free evolution Hamiltonian[35]. Similar results can be obtained from a complete path integral approach. We further note that the quantum WEP violating parameter $|r|$ cannot be arbitrarily large if one requires that the spectrum of $\hat{M}_g$ and $\hat{M}_i$ remains positive in the presence of violations. We can thus apply the perturbation theory to the test Hamiltonian of equation (8). Hereafter, we consider that the internal energy eigenstates $|1\rangle$ and $|2\rangle$ are eigenstates of $\hat{M}_i$.

For an unperturbed Hamiltonian $\hat{H}_0$ and a perturbation $\epsilon \hat{V}$, the unitary operator describing the time evolution under $\hat{\mathcal{H}} = \hat{H}_0 + \epsilon \hat{V}$ can be written as

$$\hat{U}(t) \approx e^{-\frac{i}{\hbar}\hat{H}_0(t-t_0)} - \frac{i\epsilon}{\hbar}\int_{t_0}^{t} dt_1 e^{-\frac{i}{\hbar}\hat{H}_0(t-t_1)}\hat{V}(t_1)e^{-\frac{i}{\hbar}\hat{H}_0(t_1-t_0)} \quad (9)$$

to lowest order in $\epsilon$. In the present work, $\hat{H}_0 = \hat{M}_i c^2 + \frac{p^2}{2\hat{M}_i}$ and $\epsilon \hat{V} = \left(\hat{M}_g \hat{M}_i^{-1}\right)\hat{M}_i g\hat{z}$. For a semiclassical propagation of the atoms, the time evolution along the upper path of the interferometer is thus given by

$$\hat{U}_u \approx e^{-\frac{i}{\hbar}\int_u dt \hat{H}_0} - i\left(\hat{M}_g \hat{M}_i^{-1}\right)\frac{3}{2}k_{eff}gT^2 \quad (10)$$

and similarly along the lower path

$$\hat{U}_d \approx e^{-\frac{i}{\hbar}\int_d dt \hat{H}_0} - i\left(\hat{M}_g \hat{M}_i^{-1}\right)\frac{1}{2}k_{eff}gT^2, \quad (11)$$

where $k_{eff} = 2nk$ is the effective wavevector and $n$ is the Bragg diffraction order ($n=3$ in our experiment). Finally, the dynamics of the atomic wavefunction in the interferometer is described by the evolution operator $\hat{U} = \frac{1}{2}\left(\hat{U}_u - \hat{U}_d\right)$.

In our experiment, we measure normalized atomic populations $P(\phi) = (1 - \cos\phi)/2$ in the two momentum states of the Bragg interferometer as a function of the accumulated phase $\phi$. We can count the atoms either by indistinguishably addressing them in the same detection channel on the $F=1$ and $F=2$ levels or by selectively probing them in the two hyperfine levels by using two separate channels.

When atoms are prepared in the $F=1$ or $F=2$ state at the input of the interferometer, we can measure the following transition probabilities to lowest order in $r_1$ and $r_2$:

$$\left|\langle 1|\hat{U}|1\rangle\right|^2 \approx \frac{1}{2}\left[1 - \cos\left(k_{eff}gT^2 r_1\right)\right], \quad (12)$$

$$\left|\langle 2|\hat{U}|2\rangle\right|^2 \approx \frac{1}{2}\left[1 - \cos\left(k_{eff}gT^2 r_2\right)\right]. \quad (13)$$

Atomic populations show the expected interference fringes, oscillating with a phase proportional to $a_1$ and $a_2$ of equations (4) and (5). A measurement of the differential free-fall acceleration experienced by $F=1$ and $F=2$ atoms is therefore providing a classical WEP test.

In the presence of a quantum WEP violation, the off-diagonal elements of the mass operator introduce a variation $dP$ in the atomic population that translates into a variation $d\phi = 2dP/\sin\phi$ of the interferometric phase. Therefore, to lowest order in $r_1$, $r_2$ and $|r|$, we obtain the following transition probabilities for atoms prepared in the quantum superposition state $s$ at the input of the interferometer:

$$\left|\langle 1|\hat{U}|s\rangle\right|^2 \approx \frac{1}{2}\left[\left|\langle 1|\hat{U}|1\rangle\right|^2 + 2\mathrm{Re}\left(\langle 1|\hat{U}|1\rangle e^{-i\gamma}\langle 1|\hat{U}|2\rangle^*\right)\right]$$
$$\approx \frac{1}{4}\left[1 - \cos\left(k_{eff}gT^2(r_1 + |r|\cos(\gamma + \varphi_r))\right)\right], \quad (14)$$

$$\left|\langle 2|\hat{U}|s\rangle\right|^2 \approx \frac{1}{2}\left[\left|\langle 2|\hat{U}|2\rangle\right|^2 + 2\mathrm{Re}\left(\langle 2|\hat{U}|2\rangle^* e^{-i\gamma}\langle 2|\hat{U}|1\rangle\right)\right]$$
$$\approx \frac{1}{4}\left[1 - \cos\left(k_{eff}gT^2(r_2 + |r|\cos(\gamma + \varphi_r))\right)\right], \quad (15)$$

$$\left|\langle 1|\hat{U}|s\rangle\right|^2 + \left|\langle 2|\hat{U}|s\rangle\right|^2$$
$$\approx \frac{1}{2}\left[1 - \cos\left(k_{eff}gT^2\left(\frac{r_1 + r_2}{2} + |r|\cos(\gamma + \varphi_r)\right)\right)\right]. \quad (16)$$

The phase shift accumulated by the atoms during the interferometric sequence is now showing an additional term proportional to $|r|$ (see also equation (6)). In our experiment, we cannot control $\gamma$. Indeed, both the preparation of the atomic ensemble at the input of the atom interferometer and the free evolution are imprinting phases that are randomly varying from one measurement cycle to the next introducing excess phase noise in the data. As a result, a measurement of the mean value of the interferometric phase can be used to perform a classical WEP test; at the same time, a measurement of the phase noise affecting the data provides an upper limit to the $|r|$, thus testing the quantum WEP.

**Data analysis and measurement systematics.** In our instrument, both $\gamma$ and the mechanical vibrations at the retroreflecting mirror introduce a random phase much larger than $2\pi$ that uniformly scans across the atom interference fringes. The interferometer time $T$ is kept constant ($2T = 160$ ms) during the complete measurement campaign. For each gradiometer configuration, a Lissajous figure (ellipse) is obtained by plotting the normalized population at the output ports of the upper interferometer as a function of the normalized population recorded at the lower interferometer (see Fig. 2). The differential phase is then calculated from the eccentricity and the rotation angle of the ellipse best fitting the experimental data[34]. The gravity gradient introduced by the source masses opens the ellipses thus facilitating the fitting procedure.

To evaluate the upper limit on $|r|$, we numerically generate ellipse points after introducing non-common mode phase noise between the upper and the lower cloud. The phase noise is simulated according to $6kgT^2A\cos\vartheta$ (see equation (6)), where $\vartheta$ is randomly varied between 0 and $2\pi$. Figure 4 shows parameter $A$ as function of the root mean square (RMS) noise measured on the simulated data points. In our test, we attribute all the phase noise of the measurements performed on the quantum superposition state $s$ to a violation of the quantum WEP, thus obtaining an upper limit for $|r|$. This value is found as the amplitude $A$ that provides an RMS noise of the simulated data points with respect to the best-fitting ellipse equal to the RMS value measured from the experimental data (see red line in Fig. 4).

An interesting aspect of our experiment is its robustness against the typical systematics affecting WEP tests with atom interferometers. The atomic motion is basically not perturbed by the microwave photons used to prepare the lower cloud in its internal state. As a consequence, phase shifts introduced by the Coriolis acceleration and local gravity gradients are negligible. The simultaneous operation of the Bragg interferometers on the $F=1$ and $F=2$ internal state at the magic detuning (see equation (7)) ensures equal losses towards Bragg diffraction orders other than $n=3$ and equal wavevector $k$, thus providing a high rejection ratio to losses-related systematics and to seismic noise. Finally, the implementation of the $k$-reversal measurement protocol[36] removes systematic shifts that do not depend on the direction of the Bragg lasers wavevector.

Our major sources of systematic errors arise from the second-order Zeeman shift and the AC Stark shift. In the presence of a magnetic field bias $B_0$ and a local gradient $\beta$ along the vertical direction, atoms experience an acceleration $a_m \propto \beta B_0$, corresponding to a $k$-dependent phase shift, which shows opposite sign for atoms in the $F=1$ and $F=2$ internal states. The phase shift introduced by magnetic fields has been characterized by performing measurements at different bias fields and extrapolating to zero. In the presence of intensity variations of the Bragg lasers along the propagation direction ($z$), the AC Stark effect introduces a phase shift at the $\pi$ pulse, where the spatial separation between the two interferometer arms is maximum. The sign of the shift depends on the frequency detuning and its amplitude is proportional to the spatial intensity gradients of the Bragg lasers along $z$. Intensity gradients are mainly due to the diffraction effects produced by the apertures of the optical elements needed to shape the Bragg beams. We have calculated the intensity profile of our Bragg lasers along $z$[37], averaged it over the finite size of atomic cloud, and evaluated the corresponding phase shift where the intensity gradient is maximum. In this way, we can provide an upper limit to the systematic error introduced by the AC Stark effect, which accounts for the finite size of the atomic cloud and the uncertainty of its position. We have additionally

validated our calculation by comparing the differential phase measured by the Bragg gradiometer in the $1-1$ configuration when operated with Bragg lasers red and blue detuned from the resonance by the same amount.

Finally, the differential phase measured at each gravity gradiometer is extracted from a Bayesian analysis of the experimental data. The contribution to the error budget introduced by this method depends on the knowledge of the noise power spectral density affecting the data[34]. We have estimated this contribution by generating synthetic data affected by Gaussian differential phase noise similar in magnitude (that is, RMS) to the one present in our measurements.

**Data availability.** All relevant data that support the findings of this study are available from the authors on request.

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

## Acknowledgements

This work was supported by INFN (MAGIA-Advanced experiment) and MIUR (Progetto Premiale Atom Interferometry and PRIN-2015). Č.B. is supported by the John Templeton Foundation, the Austrian Science Fund (FWF) through the Special Research Programme FoQuS, the Doctoral Programme CoQuS and Individual Project No. 24621. M.Z. is supported by the ARC Centre of Excellence for Engineered Quantum Systems (EQuS) Grant No. CE110001013 and the University of Queensland through UQ Fellowships Grant No. 2016000089.

## Author contributions

G.M.T., L.C. and G.R. conceived the idea. G.D., G.R., L.C. and F.S. performed the experiment. Č.B. and M.Z. provided the theoretical framework. M.P. contributed to the experiment and analysed the data. L.C. and G.R. wrote the paper with inputs from all the authors.

## Additional information

**Competing interests:** The authors declare no competing financial interests.

**Publisher's note**: 

