## [Peer Review File · Nature Communications]

Reviewers' comments:

Reviewer #1 (Remarks to the Author):

As I mentioned in my previous review, this is the first ever test of the quantum formulation of the equivalence principle. It is therefore a significant scientific contribution and, based on the more rigorous analysis in the new manuscript and the authors' responses, I would be happy to recommend publication in Nature Communications. Nevertheless, I still have two recommendations to improve the clarity of the presentation:

Firstly, I agree with the counterargument made by the authors, that there is no reason to a priori assume anything about potential violations. However, I think they could be clearer in stating that there could be other violations, to which they are insensitive, which do depend on the energy difference between the two levels.

Secondly, the discussion around theories which predict violation, that I suggested in my previous review would still improve the paper in my opinion. Mention of the SME and other phenomenological approaches would be sufficient, though there predictions of violations of the classical WEP in some string theories (see, e.g., Refs. 24 and 25 of the paper cited by the second Referee - Ref. 21 of the present manuscript).

Reviewer #4 (Remarks to the Author):

Overall assessment:

I think that the results in the paper are good and warrant publication in Nature Communications. The authors have conducted an impressive experiment and produced important results, which the physics research community will be very interested in. The weak equivalence principle (WEP) has not really been tested for quantum superpositions of different mass states, and so the results of Rosi et al. are an important contribution to putting bounds on possible quantum versions of the WEP. I think that the measurement of the Eotvos ratio with atoms in a superposition of internal states (and mass states) stands on its own, though the theoretical formalism put forward by the authors is also helpful in understanding it.

My main objection to the paper is that it does not adequately document the main results of the paper regarding bounds on the off-diagonal matrix elements ($|r| < 5e-8$) and the improved limits on the Eotvos ratio for the "1-2" and "1-s" measurements. Furthermore, as this paper is essentially a precision measurement of dg/g and $|r|$, the error budget provided is not sufficient and should include the magnitude of all the systematic shifts (not just the final uncertainty) that were evaluated to correct the measurements of dg/g . These are necessary for evaluating the quality of the measurement (the methods section provides a qualitative description of how to account for the shifts, but a quantitative assessment is also necessary).

Once the results (and analysis) are well documented, then the paper certainly deserves publication in Nature Communications.

The following major points should be addressed before publication proceeds

Major points to be addressed:

1. Plot or figure for $|r|$ measurement

The title of the paper claims the measurement is a “quantum test” of the equivalence principle. The quantum aspect is encapsulated in the $|r|$ measurement, so this is the most important result of the paper. Unfortunately, the main text of the paper dedicates two quasi-identical sentences to this result, and the methods section dedicates a 5-line paragraph to explain the analysis. In contrast, a large fraction of the paper is dedicated to explaining how an interferometric measurement can access r or $|r|$, in this particular case via the phase noise in the interferometer. The paper should at least have a plot that shows the phase noise (or some related quantity) that justifies the $|r| < 5e-8$ claim (along with the results of simulations). The plots of Fig. 2 and Fig. 3 are not sufficient, and I do not see how I can extract the $|r| < 5e-8$ result by visual inspection of these plots.

2. Systematics shifts in error budget

The authors present a two-order of magnitude improvement over a previous result in ref. 4. This improvement is due in large part to their impressive apparatus, which cancels out much common mode noise. However, several systematics shifts must be applied to the measurement results in order to obtain final values for the Eotvos ratio (or dg/g). After all, the ellipses in Fig. 2(a) (and in (b)) are not identical – presumably the “1-1” and “1-s” (“1-1” and “1-2”) ellipses become identical once the appropriate shifts are applied (possibly including any shifts due to gravity gradients). These systematics shifts and the means of determining them are explained qualitatively in the methods section, but any quantitative correction to the Eotvos ratio measurement should be explained quantitatively. Ideally, this would be done by detailing the magnitude of the shifts in Table 1 error budget.

3. Partially address referee #2 concern regarding measurement of phase

As referee #2 points out, if there is an off-diagonal element gravitational mass operator, then over time the $|1\rangle$ and $|2\rangle$ internal states will begin to mix (sort of “Rabi flop”). With respect to measuring “transitions between energy eigenstates”, the authors reply that they “thought about that, but we immediately abandoned the idea”. I think that a sentence or two in the text addressing this point would go a long way in showing (and contrasting) the elegance of the method actually employed in the experiment.

Minor points:

i. Eotvos ratio definition

The definition of the Eotvos ratio implies that it is always positive, yet the authors provide two measurements of the ratio (out of three) that are negative. Presumably, the absolute values can be removed in equation 1 (though the 2nd and 3rd terms are only equal to within a minus sign).

ii. Fig. 2 plot features

The “1-1” ellipses in Fig. 2(a) and 2(b) do not share the same center.

The “1-1” and “1-s” ellipses in Fig. 2(a) are substantially noisier than the ellipses in Fig. 2(b). Why is this? Is this just experimental bad luck (or perhaps good luck for the other plot). I was surprised that no comment was made in the text (or caption).

iii. Fig. 2 and Fig. 3 plot axes

The axes in these plots do not indicate what is being plotted. Presumably, what is being plotted is the

"normalized population in one of the momentum states", i.e. $P(\phi)$. However, it will be quite helpful to the readers if this indicated explicitly (even if it is mentioned in the main text).

iv. Phase range in Fig. 2 and Fig. 3

The plots in the two figures are Lissajous plots in which the output(s) of the top and bottom interferometers are plotted against each other as the phase of the interferometers is scanned (presumably by varying T , the time between the Bragg pulses). It would be helpful to know the range of the scanned phase, i.e. how many times around the ellipses does the data go?

v. What role do the masses play?

In Fig. 1, the masses are the largest objects around and appear to be quite important. In the text, the masses are barely mentioned, except for the caption of Fig. 1 which states "The external source masses are positioned to maximize the gravity gradient" and a sentence in the "gravity gradiometer" section of the "methods", which states that "their positions are accurately tuned to reach optimal conditions for the elliptical fit on the gradiometer points [28] (see below)."

From the text description, it sounds as though the masses are quite secondary and are simply used to optimize the ellipses. However, the caption indicates that the gravity gradient must be maximal. The apparatus is a gravity gradiometer, but (according to the theory presented) it is being used to search for variations in acceleration for different masses (or atomic states), not because "g" is varying in space. I have to say that I am confused by these statements (though they are not exclusive), since most of the main text (theory) seemed to indicate that a constant gravitational "g" acceleration was fine ... why does the gravity gradient need to be maximal? Furthermore, if there is a gravity gradient, and "g" is not the same for the upper and lower interferometers, then what systematic shift to the measurements was applied to compensate and correct the results (and why is it not included in Table 1)? Anyways, the role of the source masses is confusing and should be clarified.

vi. Unclear sentence (p.4, line 2-4 from top of page)

The start of the sentence "in a test theory violating WEP the ..." is awkward and should be re-written.

- Reviewer #1 (Remarks to the Authors):

As I mentioned in my previous review, this is the first ever test of the quantum formulation of the equivalence principle. It is therefore a significant scientific contribution and, based on the more rigorous analysis in the new manuscript and the authors' responses, I would be happy to recommend publication in Nature Communications. Nevertheless, I still have two recommendations to improve the clarity of the presentation:

Firstly, I agree with the counterargument made by the authors, that there is no reason to a priori assume anything about potential violations. However, I think they could be clearer in stating that there could be other violations, to which they are insensitive, which do depend on the energy difference between the two levels.

RE: We are glad that the referee finds our arguments convincing. We agree that the sensitivity of our experiment is limited by the small energy splitting.

We have added a new paragraph — at the end of the main text (page 8) — expounding what modifications to our experiment are required in order to enhance its sensitivity to WEP violations that increase with the energy difference between the superposed states and added two references.

Secondly, the discussion around theories which predict violation, that I suggested in my previous review would still improve the paper in my opinion. Mention of the SME and other phenomenological approaches would be sufficient, though their predictions of violations of the classical WEP in some string theories (see, e.g., Refs. 24 and 25 of the paper cited by the second Referee - Ref. 21 of the present manuscript).

RE: Indeed, it is very beneficial for the manuscript to highlight the broader theoretical context and ramifications of WEP tests.

We have incorporated the referee's suggestions in a new text added at the end of the first paragraph of the manuscript (page 3) and included 6 references.

- Reviewer #4 (Remarks to the Authors):

Overall assessment:

I think that the results in the paper are good and warrant publication in Nature Communications. The authors have conducted an impressive experiment and produced important results, which the physics research community will be very interested in. The weak equivalence principle (WEP) has not really been tested for quantum superpositions of different mass states, and so the results of Rosi et al. are an important contribution to putting bounds on possible quantum versions of the WEP. I think that the measurement of the Eotvos ratio with atoms in a superposition of internal states (and mass states) stands on its own, though the theoretical formalism put forward by the authors is also helpful in understanding it.

My main objection to the paper is that it does not adequately document the main results of the paper regarding bounds on the off-diagonal matrix elements ($|r| < 5e-8$) and the improved limits on the Eotvos ratio for the "1-2" and "1-s" measurements. Furthermore, as this paper is essentially a precision measurement of dg/g and $|r|$, the error budget provided is not sufficient and should include the magnitude of all the systematic shifts (not just the final uncertainty) that were evaluated to correct the measurements of dg/g . These are necessary for evaluating the quality of the measurement (the methods section provides a qualitative description of how to account for the shifts, but a quantitative assessment is also necessary).

Once the results (and analysis) are well documented, then the paper certainly deserves publication in Nature Communications.

The following major points should be addressed before publication proceeds

Major points to be addressed:

1. Plot or figure for $|r|$ measurement

The title of the paper claims the measurement is a "quantum test" of the equivalence principle. The quantum aspect is encapsulated in the $|r|$ measurement, so this is the most important result of the paper. Unfortunately, the main text of the paper dedicates two quasi-identical sentences to this result, and the methods section dedicates a 5-line paragraph to explain the analysis. In contrast, a large fraction of the paper is dedicated to explaining how an interferometric measurement can access r or $|r|$, in this particular case via the phase noise in the interferometer. The paper should at least have a plot that shows the phase noise (or some related quantity) that justifies the $|r| < 5e-8$ claim (along with the results of simulations). The plots of Fig. 2 and Fig. 3 are not sufficient, and I do not see how I can extract the $|r| < 5e-8$ result by visual inspection of these plots.

RE: We have better explained in Methods how the upper limit on $|r|$ is obtained from our measurements.

A plot showing the A parameter as a function of the RMS noise on simulated data has also been included (Fig. 4 on page 14).

2. Systematics shifts in error budget

The authors present a two-order of magnitude improvement over a previous result in ref. 4. This improvement is due in large part to their impressive apparatus, which cancels out much common mode noise. However, several systematics shifts must be applied to the measurement results in order to obtain final values for the Eotvos ratio (or dg/g). After all, the ellipses in Fig. 2(a) (and in (b)) are not identical – presumably the “1-1” and ‘1-s” (“1-1” and “1-2”) ellipses become identical once the appropriate shifts are applied (possibly including any shifts due to gravity gradients). These systematics shifts and the means of determining them are explained qualitatively in the methods section, but any quantitative correction to the Eotvos ratio measurement should be explained quantitatively. Ideally, this would be done by detailing the magnitude of the shifts in Table 1 error budget.

RE: The systematic effects evaluated in Table 1 introduce negligible bias on the differential acceleration measurements, but they contribute a significant error, more than one order of magnitude larger than our statistical uncertainty. Therefore, due to the high sensitivity of our apparatus, the differential phase angle between the two ellipses of Fig. 2 can be measured with a statistical error much smaller than the systematic uncertainty, which is presently dominating our measurement. We have included a sentence in the main text of the paper to explain that (page 6). In addition, we have better detailed in Methods how we evaluate the error contribution of the AC Stark effect (page 15-16).

3. Partially address referee #2 concern regarding measurement of phase

As referee #2 points out, if there is an off-diagonal element gravitational mass operator, then over time the $|1\rangle$ and $|2\rangle$ internal states will begin to mix (sort of “Rabi flop”). With respect to measuring “transitions between energy eigenstates”, the authors reply that they “thought about that, but we immediately abandoned the idea”. I think that a sentence or two in the text addressing this point would go a long way in showing (and contrasting) the elegance of the method actually employed in the experiment.

RE: Following the recommendation of the referee, we have addressed the issue in the main text of the paper (page 4, after Eq. 3), highlighting the drawbacks of an experiment that would directly detect the population transfer between the two hyperfine levels.

Minor points:

i. Eotvos ratio definition

The definition of the Eotvos ratio implies that it is always positive, yet the authors provide two measurements of the ratio (out of three) that are negative. Presumably, the absolute values can be removed in equation 1 (though the 2nd and 3rd terms are only equal to within a minus sign).

RE: We have corrected the measurement results (page 6 and 7) and removed the minus signs.

ii. Fig. 2 plot features

The “1-1” ellipses in Fig. 2(a) and 2(b) do not share the same center.

The “1-1” and “1-s” ellipses in Fig. 2(a) are substantially noisier than the ellipses in Fig. 2(b). Why is this? Is this just experimental bad luck (or perhaps good luck for the other plot). I was surprised that no comment was made in the text (or caption).

RE: Figure 2 has been corrected and the ellipses have been re-centred. The relative position of the ellipses centres is due to biases on the two detection channels that are unimportant to the differential acceleration measurement.

Regarding the noise level, the experiment was simply more stable in the second run (~ 1.5). As an example, night-time runs are always more stable than day-time ones. We prefer not to discuss this point in the article as it is not important for the measurement itself.

Concerning the loss of contrast observed on the measurements involving the interferometer on the $|2\tilde{n}\rangle$ state, we attribute it to the de-focusing effect induced on the atoms by the blue-detuned Bragg lasers. We have included a sentence in the caption of Fig. 2 to explain that.

iii. Fig. 2 and Fig. 3 plot axes

The axes in these plots do not indicate what is being plotted. Presumably, what is being plotted is the “normalized population in one of the momentum states”, i.e. $P(\phi)$. However, it will be quite helpful to the readers if this indicated explicitly (even if it is mentioned in the main text).

RE: The labels have been corrected in both Fig. 2 and 3.

iv. Phase range in Fig. 2 and Fig. 3

The plots in the two figures are Lissajous plots in which the output(s) of the top and bottom interferometers are plotted against each other as the phase of the interferometers is scanned (presumably by varying T , the time between the Bragg pulses). It would be helpful to know the range of the scanned phase, i.e. how many times around the ellipses does the data go?

RE: In our setup, both g (see page 4 and 13) and the mechanical vibrations at the retro-reflecting mirror introduce a random phase term much larger than 2ϕ that uniformly scans across atom interference fringes. The interferometer time T is kept constant ($2T = 160$ ms).

A sentence has been added in the Methods (page 14), at the beginning of the paragraph on “Data analysis and measurement systematics”.

v. What role do the masses play?

In Fig. 1, the masses are the largest objects around and appear to be quite important. In the text, the masses are barely mentioned, except for the caption of Fig. 1 which states “The external source masses are positioned to maximize the gravity gradient” and a sentence in the “gravity gradiometer” section of the “methods”, which states that “their positions are accurately tuned to reach optimal conditions for the elliptical fit on the gradiometer points [28] (see below).” From the text description, it sounds as though the masses are quite secondary and are simply used to optimize the ellipses. However, the caption indicates that the gravity gradient must be maximal. The apparatus is a gravity gradiometer, but (according to the theory presented) it is being used to search for variations in acceleration for different masses (or atomic states), not because “ g ” is varying in space. I have to say that I am confused by these statements (though they are not exclusive), since most of the main text (theory) seemed to indicate that a constant gravitational “ g ” acceleration was fine ... why does the gravity gradient need to be maximal? Furthermore, if there is a gravity gradient, and “ g ” is not the same for the upper and lower

interferometers, then what systematic shift to the measurements was applied to compensate and correct the results (and why is it not included in Table 1)? Anyways, the role of the source masses is confusing and should be clarified.

RE: The positions of the source masses are accurately tuned to maximize the gravity gradient. In this way, it is possible to open the ellipses and reach optimal conditions for the elliptical fit on the gradiometer data points. We have modified the sentence both in the caption of Fig. 1 and in the gravity gradiometer section of Methods (page 10) to clarify this point. A sentence has also been added at the end of the first paragraph of the “Data analysis and measurement systematics” section (page 14).

With respect to the question “Furthermore, if there is a gravity gradient, and “g” is not the same for the upper and lower interferometers, then what systematic shift to the measurements was applied to compensate and correct the results?”. Please, note that we obtain Eötvös parameters by comparing accelerations of atoms in the lower interferometer for different internal states - see second sentence below eq. (5). The upper cloud only acts as a common reference: we do not take differential acceleration between atoms in the lower and in the upper cloud to bound WEP violations.

vi. Unclear sentence (p.4, line 2-4 from top of page)

The start of the sentence “in a test theory violating WEP the ...” is awkward and should be re-written.

RE: Thanks for pointing that out. The sentence has been rephrased.

Additional changes to the manuscript:

- 1. We have replaced the sentence “where $\cos\vartheta$ is randomly varied between -1 and $+1$ ” of page 14 with “where ϑ . Indeed, the previous formulation was misleading.*
- 2. We have been more specific in the last sentence of page 16, detailing the kind of noise used in the simulation.*
- 3. We have corrected some misprints in the bibliography.*

REVIEWERS' COMMENTS:

Reviewer #1 (Remarks to the Author):

Since the previous round of review, the authors have made several improvements to the manuscript, following both my and the other Referee's suggestions. In particular, they have significantly expanded on their quantitative error analysis. I am now happy to recommend publication in Nature Communications without further modification.

Reviewer #4 (Remarks to the Author):

The changes are good and sufficient. The added text is quite helpful in making the paper more understandable and accessible. The added figure 4 is helpful (though I had hoped to see a figure with the noise on the ellipse fits, i.e. the residuals) ... though the caption for this figure is a bit short. The discussion of the sources of systematic uncertainty is also improved and clarifies a concern regarding systematic shifts (which are apparently quite small).

All in all, I recommend publication.

REPLY TO REVIEWERS' COMMENTS:

Reviewer #1 (Remarks to the Author):

Since the previous round of review, the authors have made several improvements to the manuscript, following both my and the other Referee's suggestions. In particular, they have significantly expanded on their quantitative error analysis. I am now happy to recommend publication in Nature Communications without further modification.

Reviewer #4 (Remarks to the Author):

The changes are good and sufficient. The added text is quite helpful in making the paper more understandable and accessible. The added figure 4 is helpful (though I had hoped to see a figure with the noise on the ellipse fits, i.e. the residuals) ... though the caption for this figure is a bit short. The discussion of the sources of systematic uncertainty is also improved and clarifies a concern regarding systematic shifts (which are apparently quite small).

All in all, I recommend publication.

Re: We thank the reviewers for their comments and their constructive criticism, which significantly helped in improving our paper. Following the suggestion by Reviewer #4, we have expanded the caption of Figure 4 and added a sentence to explain how the error bars have been obtained.